# Estradiol Upregulates the Expression of the TGF-β Receptors *ALK5* and *BMPR2* during the Gonadal Development of *Schizothorax prenanti*

**DOI:** 10.3390/ani11051365

**Published:** 2021-05-11

**Authors:** Taiming Yan, Songpei Zhang, Yueping Cai, Zhijun Ma, Jiayang He, Qian Zhang, Faqiang Deng, Lijuan Ye, Hongjun Chen, Liang He, Jie Luo, Deying Yang, Zhi He

**Affiliations:** College of Animal Science and Technology, Sichuan Agricultural University, Chengdu 611130, China; yantaiming@sicau.edu.cn (T.Y.); spzhang2428@163.com (S.Z.); caiypfishcounts@163.com (Y.C.); 2018202029@stu.sicau.edu.cn (Z.M.); cnhjykbcg@163.com (J.H.); s20175403@stu.sicau.edu.cn (Q.Z.); s20175304@stu.sicau.edu.cn (F.D.); lijuanye123@163.com (L.Y.); jet4rhzd1@163.com (H.C.); heliang930934@163.com (L.H.); luojiexs@163.com (J.L.); Deyingyang@sicau.edu.cn (D.Y.)

**Keywords:** location, expression pattern, immunoreactivity, E_2_

## Abstract

**Simple Summary:**

*Schizothorax prenanti*, known as the ya-fish, is mainly distributed in regions adjacent to the Qinghai-Tibet Plateau (QTP) and is an endemic fish species with great economic importance in aquaculture in Western China. In the present study, we were aimed to explore the functions of *ALK5* and *BMPR2* during the gonadal development of *S. prenanti*. Our results suggest that *ALK5* and *BMPR2* may play a potentially vital role in both folliculogenesis and spermatogenesis in *S. prenanti*.

**Abstract:**

TGF-β receptors play important roles in mediating TGF-β signals during gonadal development. To identify the functions of TGF-β receptors, including the type I receptor (activin receptor-like kinase 5, *ALK5*) and type II receptor (bone morphogenetic protein receptor 2, BMPR2), during the gonadal development of *S. prenanti*, the full-length cDNA sequences of *ALK5* and *BMPR2* were isolated and characterized. Their expression patterns in developing gonads and in the gonads of exogenous estradiol (E_2_) -fed fish were analyzed. The cDNAs of *ALK5* and *BMPR2* were 1925 bp and 3704 bp in length and encoded 501 and 1070 amino acid residues, respectively. *ALK5* and *BMPR2* were mostly expressed in gonads, particularly in cortical alveoli stage ovaries and mid-spermatogenic stage testes; however, the overall level of *BMPR2* mRNA was higher than that of *ALK5* during gonadal development. Furthermore, immunohistochemical signals of ALK5 and BMPR2 were mostly detected at chromatin nucleolar oocytes and perinuclear oocytes in ovaries and at spermatocytes and spermatogonia in testes. Exogenous E_2_ induces the gonadal expression of *ALK5* and *BMPR2*, and *BMPR2* is more responsive to E_2_ than *ALK5*. These results suggest that ALK5 and BMPR2 might play a potentially vital role in both folliculogenesis and spermatogenesis in *S. prenanti*.

## 1. Introduction

Transforming growth factor-beta (TGF-β) superfamily receptors, including the type I receptor (activin receptor-like kinase 5, ALK5) and type II receptor (bone morphogenetic protein receptor 2, BMPR2), are key factors in the transmission of signals that regulate gonadal development [1,2,3]. As a high-affinity signaling complex, BMPR2 and ALK5 transmit intracellular signals through the phosphorylation of downstream effectors during the reproductive process by stimulating and subsequently activating TGF-βs [2,4,5,6,7].

*ALK5* is mainly expressed in mammalian gonads and expressed at high levels in cattle ovaries [8]. *ALK5* has also been detected in human germ cells [9] and in oocytes and granulosa cells of mice and pigs at different developmental stages [10]. During the development of the corpus luteum (CL) from early to late CL, the expression of *ALK5* continually increases [11]. In fish, *ALK5*, which is abundant in gonads and muscles, is distributed in the tissues of *Oncorhynchus mykiss* [12] and mainly expressed in the liver of *Oreochromis niloticus* [13] and *Carassius gibelio* [14]. These results indicated that the expression of *ALK5* varies in different species.

Mammalian *BMPR2* is widely expressed in various tissues, particularly in porcine oocytes [15] and granulosa cells of ewes [3]. The expression of *BMPR2* in chicken exhibits a gradual increase during follicular development [16]. In addition, the expression of *BMPR2* in pigs is higher at both the embryonic stage and pre- and late puberty than at other stages [17], but its expression is stable from early to late CL [11]. In fish, *BMPR2* is widely distributed in *Danio rerio* and is mainly expressed in the testis and liver [18]. However, the abundant expression has been detected in the brain and liver of *C. gibelio* [14]. These results show that *BMPR2* is expressed in different tissues and species, suggesting that *BMPR2* plays different roles in various species.

ALK5 promotes the development of germ cells [9], oocytes and granulosa cells [10] in humans, inhibits pig granulosa cells apoptosis [19], and participates in establishing luteolysis in mare steroidogenic cells [11], and a lack of ALK5 induces severe defects in mice yolk sac development [20]. BMPR2 participates in embryonic development of porcine [17], follicular development of chicken [16] and cumulus cells expansion, proliferation and apoptosis of bovine [21] and maintains steroidogenesis in human [22], progesterone biosynthesis in rat granulosa cells [23], and ovarian cells functional homeostasis in hamster [24]. There is no doubt that ALK5 and BMPR2 are essential and vital factors and participate in these biological processes by enhancing or abating upstream signals. Therefore, these results are meaningful for obtaining further clarification of their roles in signal transduction.

*Schizothorax prenanti*, which belongs to the subfamily Schizothoracinae of the Cyprinidae family, is mainly distributed in regions adjacent to the Qinghai-Tibet Plateau (QTP) and an endemic fish species with great economic importance in aquaculture in Western China [25]. Our previous report showed that GDF9 plays an important role in folliculogenesis and spermatogenesis in *S. prenanti* [26], but the downstream signal transduction mechanism remains unclear. Because ALK5 and BMPR2 are important and necessary for signal transduction, their regulatory roles in the gonadal development process of *S. prenanti* should be unraveled. In the present study, we cloned the *ALK5* and *BMPR2* cDNA sequences, analyzed the expression patterns of *ALK5* and *BMPR2* at both the mRNA and protein levels, and detected their expression in juvenile fish gonads via the withdrawal of estradiol (E_2_) administration. All these results allow in-depth exploration of the functions of *ALK5* and *BMPR2* during the gonadal development of *S. prenanti*.

## 2. Materials and Methods

### 2.1. Animals

Adult *S. prenanti* (body length, 38.73 ± 3.52 cm; body weight, 1194.79 ± 197.86 g) were purchased from the Lu Shan Sword Fishing Company (Sichuan, China). All these fish were maintained in 60.0 × 50.0 × 40.0 cm^3^ indoor tanks at the Aquaculture Laboratory of Sichuan Agricultural University (Chengdu, China) under a natural photoperiod (16 h light: 8 h dark) at a temperature of 19.0 ± 0.1 °C. Fish were supplied with a flow of the freshwater system. All the experiments were performed following the Animal Research and Ethics Committees of Sichuan Agricultural University and followed the animal experiments guidelines of Sichuan Agricultural University (Approval No: 20170031).

### 2.2. Tissue Sampling

All adult fish were anesthetized with 0.02% tricaine buffer (80 µg/L) (Sigma, St. Louis, MO, USA) for 10 min after a 24 h fast. Tissues (n = 5), including the hypothalamus, telencephalon, epithalamus, heart, kidney, blood, liver, eyes, muscle, spleen, swim bladder, intestine, head kidney, medulla oblongata, skin, pituitary, gills, testes, and ovaries, were collected. Only the gonads were collected from the remaining individuals (n = 45). The tissues were snap-frozen in liquid nitrogen and stored at −80 °C until use. Fresh gonad samples were immobilized in Bouin’s solution and stored in 75% ethanol.

According to previous reports [25,27], the gonadal developmental stages of fish were identified through histological analysis of the gonad sections (the 5 μm sections were stained with hematoxylin–eosin). The gonads were classified as follows: ovaries at the chromatin nucleolar stage (CNS), perinucleolar stage (PS), cortical alveoli stage (CAS), mid-vitellogenic stage (MVS), and late vitellogenic stage (LVS) and testes at the spermatogonium stage (SGT), early spermatogenic stage (EST), mid-spermatogenic stage (MST) and late spermatogenic stage (LST).

### 2.3. Total RNA Isolation and Sequence Cloning

Total RNA was extracted using the TRIzol reagent (Invitrogen, Carlsbad, CA, USA) according to the manufacturer’s instructions. The integrity of total RNA was assessed by 1% agarose gel electrophoresis, and the optical density absorption ratios at 260 and 280 nm were quantified with a photometer (Bio-Rad, Hercules, CA, USA). Before reverse transcription, total RNA was enzymatically digested with gDNA Eraser (TaKaRa, Dalian, China).

The ovarian cDNAs were transcribed from 1.0 μg of total RNA from the ovary using a BD SMART™ PCR cDNA synthesis kit (Invitrogen, Carlsbad, CA, USA) according to the manufacturer’s instructions. Degenerate primers were designed according to the conserved regions of *ALK5* and *BMPR2*. The 5′-RACE and 3′-RACE primers were designed based on their partial sequences (Table 1). Except for the first round of annealing (performed at 54 °C) and the second round of annealing (performed at 56 °C), the PCR methods and procedures were performed as described previously. The PCR mixtures comprised 5.0 μL of 2× Taq buffer (DSBIO, Guangzhou, China), forward and reverse primers (0.5 µM each), 3 μL of ddH_2_O, and 40 ng of cDNA, and the reactions were performed with the following program: pre-denaturation at 94 °C for 3 min, 30 cycles of denaturation at 94 °C for 30 s, annealing at 54 °C for 30 s, and extension at 72 °C for 90 s and elongation at 72 °C for 10 min. The PCR products were purified from a 1.5% agarose gel using a universal DNA purification kit (TIANGEN, Beijing, China), ligated into the pMD-19T vector (TaKaRa, Dalian, China), and then introduced into *E. coli* DH5α competent cells (Takara, Dalian, China). The positive clone was isolated and sequenced by Chengdu Tsingke Biotechnology Co., Ltd. (Chengdu, China).

The *ALK5* and *BMPR2* amino acid sequences of *S. prenanti* and other vertebrates were aligned by BLAST (http://bLast.ncbi.nLm.nih.gov/BLast.cgi, accessed on 24 September 2020). The phylogenetic tree was constructed using the neighbor-joining algorithm of MEGA 7.0. The reliability of the analysis was assessed by 1000 bootstrap replicates.

### 2.4. Real-Time Quantitative PCR Analysis

According to the manufacturer’s recommended protocol, the first-strand cDNA synthesis kit (Thermo, Waltham, MA, USA) was used for cDNA synthesis, and the cDNA quality was verified by the stable amplification of *β-actin*. As shown in our previous study, we used both *β-actin* and 18S *r*RNA as reference genes for controlling the error between samples in the analysis of *ALK5* and *BMPR2* expression. For all standard curves, the primer amplification efficiencies of the genes were 94.7–96.8% and 0.990 < *R*^2^ < 0.999. The target genes were normalized to the reference genes (geometric averaging of the *β-actin* and 18S *r*RNA Ct values), and the expression levels were compared using the relative Ct method.

### 2.5. Western Blotting Analysis

Primary antisera of BMPR2 (LS-C178875) and ALK5 (LS-C312882) were purchased from LifeSpan BioSciences (Seattle, WA, USA). Western blot analysis was performed according to the standard protocol. The ovarian and testicular homogenates were separated on a 12% SDS–PAGE gel, and the proteins were then transferred to PVDF (methanol-activated polyvinylidene difluoride) membranes (Merck Millipore, Darmstadt, Germany) using Bio-Rad trans-blot (Bio-Rad, Hercules, CA, USA). The membranes were incubated overnight in 5% nonfat milk powder in 10 mM PBST buffer at 4 °C, blocked and incubated again with the primary antiserum (1:800 dilution) diluted with 5% nonfat milk powder in 10 mM PBST, at 25 °C for 2 h. The membrane was subjected to three 5 min washes with PBST and incubated in horseradish peroxidase (HRP)-conjugated goat anti-rabbit IgG (Boster, Wuhan, China; 1:500 dilution) diluted with 5% nonfat milk powder in 10 mM PBST for 1 h at 25 °C. According to the protocol recommended by the manufacturer of the ECL kit (MXB, Fuzhou, China), the membranes were washed with PBST and detected using Bio-Rad ChemiDoc^TM^ MP (Bio-Rad, Hercules, CA, USA).

### 2.6. Immunohistochemistry Analysis

In 3% hydrogen peroxide, 5 μm-thick paraffin sections of the gonads were deparaffinized, dehydrated and incubated for 30 min. The sections were subjected to three 5 min washes with phosphate-buffered saline (PBS, 10 mM) and incubated with 10% normal goat serum (Boster, Wuhan, China) for 20 min to block nonspecific reactions. The sections were then incubated with rabbit BMPR2 and ALK5 (1:300 dilution) at 25 °C for 2 h, washed three times and incubated with a secondary antibody [horseradish peroxidase (HRP)-conjugated goat anti-rabbit IgG; 1:1000 dilution]. After rinsing with PBS, the sections were developed with an ECL reagent kit, mounted, examined under a microscope, and digitally photographed (Nikon, Tokyo, Japan).

### 2.7. Effect of ALK5 and BMPR2 Expression on E2-Fed Gonads

The fish were hatched in a laboratory at Sichuan Agricultural University (Chengdu, China) and supplied with continuously flowing fresh water at 19.0 ± 1.0 °C and natural light. Four hundred fifty fish (46.50 ± 1.55 mm) were randomly assigned to 15 indoor fiberglass tanks (60.0 × 50.0 × 40.0 cm^3^, 30 fish per tank) in the laboratory.

The feeding trial involved five diets: the blank and alcohol control diets and E_2_-supplemented diets (50, 100 and 150 mg/kg E_2_). Before being added to the diets, E_2_ (Sigma, St. Louis, MO, USA) was dissolved in alcohol. The fish in the alcohol control diet-fed group were treated like those in the E_2_ diet-fed groups. The fish belonging to the control group were fed a common diet throughout the experiment, which lasted 4 weeks. All fish were fed a common diet for 2 weeks before initiating the experiment to allow their acclimatization to the experimental diet and conditions. All fish were fed twice a day at 8:00 and 18:00, and the proportion was 3% of the body weight. The residual diets were removed with a siphon tubular. The survival ratio of every group was calculated daily.

Ten fish from each group were anesthetized via a tricaine buffer bath (80 µg/L), and gonad samples from each of these fish were collected and stored at −20 °C in RNAlater (Takara, Dalian, China). Total RNA extraction, cDNA synthesis, and target gene expression analysis were performed as described above.

### 2.8. Data Analysis

All the data are presented as the means ± SEMs and subjected to one-way analysis of variance (ANOVA) followed by Duncan’s multiple range test to determine the significance of the differences among the treatment groups or tissues at the level of *p* < 0.05 after the Kolmogorov–Smirnov test. If it does not fit the normality and homoscedasticity distribution, the data were used the mathematical transformation (logarithmic 10 + 10) to normalize the data before statistical analysis. The statistical analyses were performed using SPSS 21.0 software (SPSS Inc., Chicago, IL, USA).

## 3. Results

### 3.1. Nucleotide and Deduced Amino Acid Sequences

The full-length *S. prenanti ALK5* cDNA sequence consisted of 1958 base pairs (bp), including a 62-bp 5′UTR and a 356-bp 3′UTR. The putative 501 amino acid residues contained two N-glycation sites, a transmembrane domain (Ala^123^-Ile^145^), a typical SGSGSG domain (Thr^173^-Lys^230^) and an S-TKc domain (Ile^203^-Leu^490^). The *BMPR2* cDNA sequence of *S. prenanti* consisted of 3704 bp and encoded 1070 amino acids, including six N-glycosylation sites, one transmembrane domain (Ala^157^-Tyr^179^) and one STYKc (Lys^210^-Ser^580^) domain, and some low-complexity domains were deduced among the putative *BMPR2* amino acid sequence. The phylogenetic analysis showed that *ALK5* and *BMPR2* of *S. prenanti* converged with those of fish and exhibited the highest identity with those of *C. auratus* and *D. rerio*, respectively (Figure 1).

### 3.2. Tissue Distribution of ALK5 and BMPR2

The tissue expression patterns of *ALK5* and *BMPR2* mRNA in adult females and males were analyzed by qRT–PCR. The *ALK5* and *BMPR2* mRNA expression levels in gonads were significantly higher than that in non-gonadal tissues (*p* < 0.05) (Figure 2A,C), and the ovarian expression levels were higher than those in the testes. Interestingly, the expression of *BMPR2* in gonads was higher than that of *ALK5*.

Western blot analysis of tissue homogenates from the ovary and testis only detected specific bands of approximately 50 kDa and 75 kDa (Figure 2B,D). No bands were observed when the primary antiserum was replaced with PBS (Figure 2B,D).

ALK5 and BMPR2 protein bands with molecular weights of 50 kDa and 75 kDa were only detected in the ovary and testis. Among these bands, the size of BMPR2 was similar to that of the band sheared into the cytolemma.

### 3.3. Expression Levels of ALK5 and BMPR2 mRNAs during Gonad Development

Similarly, the expression level of *BMPR2* was higher than that of *ALK5*. During ovarian development, *ALK5* and *BMPR2* expression continued to increase until peaking at the CAS and then decreased (*p* < 0.05) (Figure 3A,C). However, some differences were detected during the testis development process. During testis development, the expression of *BMPR2* increased gradually until reaching its maximum level at the MST and then decreased to its minimum at the LST (Figure 3D). *ALK5* was higher at the SGT and MST and reached its minimum level at the EST and LST (Figure 3B).

### 3.4. Localization of ALK5 and BMPR2 during Gonadal Development

Immunohistochemical signals of ALK5 were detected in both the ovaries and testis of *S. prenanti*. Specifically, positive signals were detected in the cytoplasm of chromatin nucleolar oocytes and perinucleolar oocytes (Figure 4A,B) and the cytolemma and granulosa cells of cortical alveoli oocytes and mid-vitellogenic oocytes (Figure 4B–D). Strong signals were also detected in the cytoplasm of chromatin nucleolar oocytes and perinucleolar oocytes (Figure 4A) as well as in spermatogonia, spermatocytes, spermatids and Sertoli cells (Figure 4E). When PBS was used instead of the primary antiserum, no positive signals were observed in the gonads (Figure 4F).

In the ovary, BMPR2-positive signals were found in the cytoplasm of chromatin nucleolar oocytes and perinucleolar oocytes (Figure 4G). Signals were also detected in the cytodeme and granulosa cells of cortical alveoli oocytes and mid-vitellogenic oocytes (Figure 4H–J). Moreover, strong signals were detected in the cytolemma and cytoplasm of cortical alveoli oocytes (Figure 4H,I). In the testis, positive signals were detected in the cytoplasm of the spermatogonia, spermatocyte spermatids and Sertoli cells (Figure 4K). No positive signals were observed when the primary antiserum was replaced with PBS (Figure 4L).

### 3.5. Expression of ALK5 and BMPR2 Induced by E_2_

The survival rate, which was higher than 95% in all the groups, did not show significant differences during the experimental period (*p* > 0.05). As the E_2_ concentration and treatment time increased, the expression of *ALK5* and *BMPR2* gradually increased in all the treatment groups. In addition, *ALK5* and *BMPR2* expression in the group treated with 200 mg/kg E_2_ was higher than in the other groups (*p* < 0.05). Moreover, the expression of both *ALK5* and *BMPR2* was induced by E_2_, even though no significant difference was detected among the treatment groups on some weeks. All of these results showed that the expression of *ALK5* and *BMPR2* was induced by E_2_ and that the expression of *BMPR2* is higher than that of *ALK5* (Figure 5A,B).

## 4. Discussion

In the present study, the *ALK5* and *BMPR2* cDNA sequences of *S. prenanti* were characterized. Comparing the amino acid sequences showed that these *ALK5* and *BMPR2* cDNA sequences exhibited high homogeneity to those of other fish species. Both *ALK5* and *BMPR2* contained a conserved transmembrane domain, which is essential and plays a pivotal role in ligand activation [28,29]. Despite the divergence in their sequence, the serine/threonine kinase domains, which play a crucial role in ligand recognition, are conserved in the *ALK5* and *BMPR2* amino acid sequences, similar to the results observed in buffalo [8] and *O. mykiss* [12]. All of the results reveal the relatively high homogeneity of the different kinase domains and sequences with those of other vertebrates, suggesting that *ALK5* and *BMPR2* play conserved roles in signal transduction [13].

The expression of *ALK5* and *BMPR2* was detected in almost all tissues, particularly in gonads. Similarly, *ALK5* is expressed in all tissues of *O. mykiss* [12] and is expressed at high levels in the buffalo ovary [8], the liver of *O. niloticus* [13] and *C. gibelio* follicle cells [14]. During oocytes development, the expression of *ALK5* in ovaries at the early vitellogenic stage is higher than that at other stages [14]. *ALK5* is more highly expressed in luteinized than in nonluteinized follicular cells of marmosets [30]. *ALK5* expression in bovine granulosa cells is significantly increased after 12 h of treatment but decreased after 24 h of treatment [31]. In the mouse testes, *ALK5* is involved in developing spermatogonia [32]. Although the expression of TGF-β and *ALK5* is decreased in spermatogenesis [1], abundant *ALK5* mRNA levels are detected during puberty, which confirms that *ALK5* plays an irreplaceable role during this process [32,33]. Therefore, these results indicate that *ALK5* is essential for oocytes growth and spermatogenesis.

Although it is not specific to non-gonadal tissues, *BMPR2* plays a pivotal role during gonadal development. In *C. gibelio*, *BMPR2a* mRNA is mainly expressed in the brain, and *BMPR2b* is expressed in the liver [14]. However, in *D. rerio*, although expression of *BMPR2a* and *BMPR2b* mRNAs is universal, these mRNAs are mainly expressed in the testis and liver [18]. Importantly, *BMPR2* gradually increases during oocytes and follicular development [16]. In piglets [34] and adult pigs [35,36], *BMPR2* is significantly expressed in follicles and oocytes at various stages. Some differences have been detected in rats: *BMPR2* has been detected in oocytes and granulosa cells at all stages, except for primitive follicular granulosa cells [37]. In contrast, *BMPR2* is expressed at higher levels in primary oocytes than at other stages [14]. Moreover, *BMPR2* expression decreases as germ cells enter the meiosis stage, but *BMPR2* expression was detected at all stages of testicular development [38]. During mouse testes development, *BMPR2* expression at the early developmental stage is significantly higher than that in mature testes [39]. The wide distribution of *BMPR2* in gonadal and nongonadal tissues may involve a vast array of biological responses and functions.

The locations of ALK5 and BMPR2 further explain their functions in ovarian and testicular development. ALK5 and BMPR2 were detected in mouse oocytes and granulosa cells at all stages, similar to the results obtained with rats and pigs [10,37,40]. ALK5-positive signals have been detected in granulosa cells [41], and BMPR2 is detected in follicles at different developmental stages [17]. Moreover, ALK5 and BMPR2 are expressed in cattle and sheep oocytes [42] and in human primitive and primary follicles [10]. Some studies have shown that ALK5 and BMPR2 are more abundantly expressed at Mid-CL than at other stages [11] and that ALK5 and BMPR2 are expressed in both granulosa cells and theca cells of piglets [43]. ALK5 and BMPR2 are essential in inhibiting bovine granulosa cell apoptosis [19] and affecting cumulus cell expansion and proliferation [21]. All of these results demonstrate that ALK5 and BMPR2 might be closely related to regulating germ cells and somatic cell development.

The present study showed that the expression of *ALK5* and *BMPR2* was induced by E_2_ during *S. prenanti* gonad development, and the expression of *BMPR2* was higher than that of *ALK5*. In addition, the *ALK5* and *BMPR2* mRNA levels generally increased with increases in the treatment time and dose. The expression of *ALK5* and *BMPR2* was particularly promoted in bovine granular cells by low and high concentrations of E_2_ [42]. Consistently, *ALK5* and *BMPR2* expression in ovine granular cells is significantly increased by daidzein, similar to the results obtained with E_2_ [44]. Analogously, treatment with FSH and E_2_ for different times and at different doses also promotes *ALK5* and *BMPR2* expression in granulosa cells of bovine [45]. Recently, some researchers have found that neonatal exposure to flutamide and 4-tertoctylphenol (OP, an estrogenic compound) can upregulate *ALK5* and *BMPR2* expression in small antral follicles of pigs [43]. Therefore, we hypothesized that it could result in differential activation of signaling pathways and an alteration of the gene expression profile in gonads. These results further confirmed that *ALK5* and *BMPR2* might be the key signaling factors during gonadal development and depend on precise regulation of the stages and hormone levels.

Interestingly, the fold change in *BMPR2* expression obtained in this study was higher than that in *ALK5* expression. Similarly, the expression of *BMPR2* and *ALK5* in ovarian follicles is stimulated by E_2_, and the stimulated expression of *BMPR2* is clearly higher than that of *ALK5* [42]. In addition, the expression of *BMPR2* in bovine granular cells is higher than that of *ALK5* [42,45]. In general, seven type I receptors, also known as activin receptor-like kinases (ALKs, ALK1–7), have been reported to date. TGF-β provides signals in most cells by binding to TGF-βRII and thus forming a complex with type I receptors. Moreover, some studies have found that TGF-β signaling is dependent on ALK5, and BMPR2 is not the only type I receptor [46,47,48]. In addition to activating ALK5 and BMPR2, as a prerequisite of many TGF-β and BMP-signaling pathways, participates in other signaling transduction pathways [4,5,6,47].

## 5. Conclusions

In summary, *ALK5* and *BMPR2* were characterized, and their mRNAs were prominently expressed in the ovary and testis, particularly at the CAS and MST. ALK5 and BMPR2 signals were observed in the cytoplasm of oocytes, spermatocytes, spermatids and somatic cells. Exogenous E_2_ promoted the expression of *S. prenanti ALK5* and *BMPR2*, and the expression of *BMPR2* was higher than that of *ALK5*. Combined with the findings from our previous study, the results obtained in the present study suggest that GDF9 signals transmitted via ALK5 and BMPR2 might participate in folliculogenesis and spermatogenesis in *S. prenanti*.

## Figures and Tables

**Figure 1 animals-11-01365-f001:**
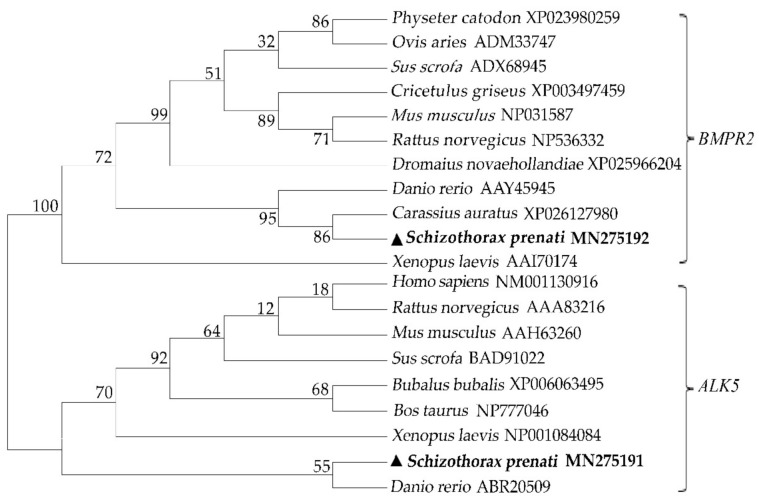
Phylogenetic analysis of *ALK5* and *BMPR2* with those of other vertebrates. The *ALK5* and *BMPR2* sequences of the other vertebrates were downloaded from Entrez (NCBI). The phylogenetic tree was analyzed using the neighbor-joining algorithm of Mega 7.0. The numbers at the forks are the bootstrap proportions. *S. prenanti* is marked with black triangles.

**Figure 2 animals-11-01365-f002:**
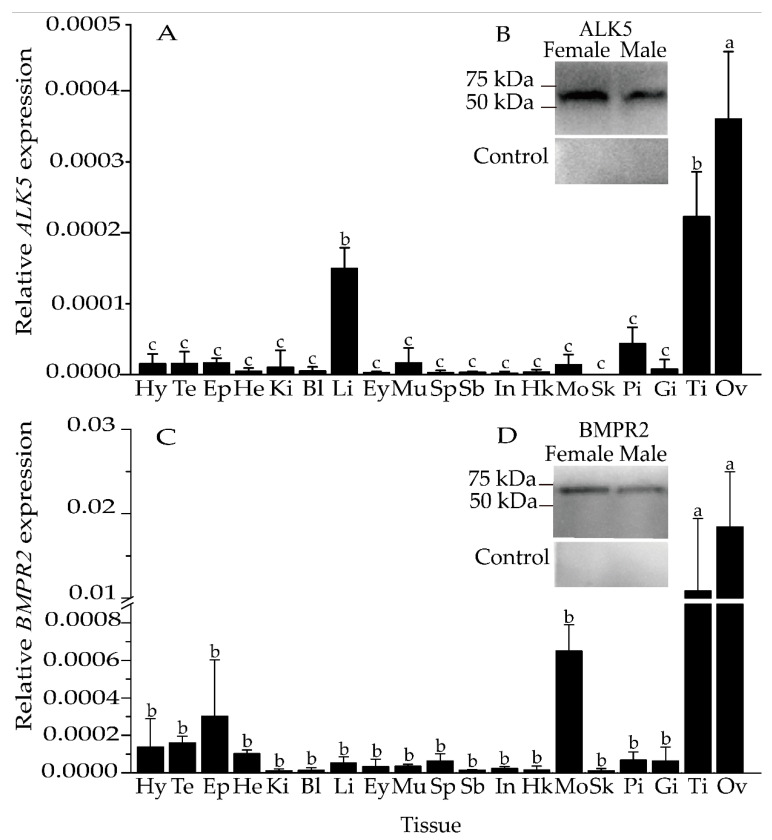
Tissue distribution of *ALK5* (**A**) and *BMPR2* (**B**) in *S. prenanti* determined by qRT–PCR, and ALK5 and BMPR2 immunoreactivity in the ovary and testis (**C**,**D**). Hy, hypothalamus; Te, telencephalon; Ep, epithalamus; He, heart; Ki, kidney; Bl, blood; Li, liver; Ey, eye; Mu, muscle; Sp, spleen; Sb, swim bladder; In, intestine; Hk, head kidney; Mo, medulla oblongata; Sk, skin; Pi, pituitary; Gi, gill; Ti, testis; Ov, ovary. The data are represented as the means ± SEMs (n = 5). Means marked with different letters are significantly different (*p* < 0.05).

**Figure 3 animals-11-01365-f003:**
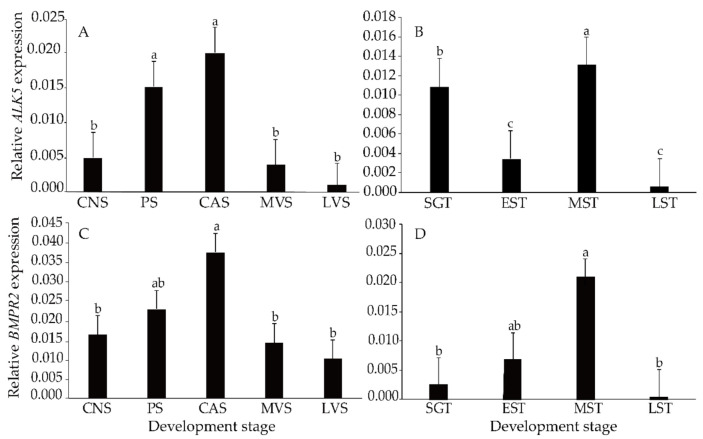
Expression of *ALK5* (**A**,**B**) and *BMPR2* (**C**,**D**) during gonadal development (n = 5). CNS, ovary at the chromatin nucleolar stage; PS, ovary at the perinuclear stage; CAS, ovary at the cortical alveoli stage; MVS, ovary at the mid-vitellogenic stage; LVS, ovary at the late vitellogenic stage; SGT, testis at the spermatogonium stage; EST, testis at the early spermatogenic stage; MST, testos at the mid-spermatogenic stage; LST, testis at the late spermatogenic stage. Means marked with different letters are significantly different (*p* < 0.05).

**Figure 4 animals-11-01365-f004:**
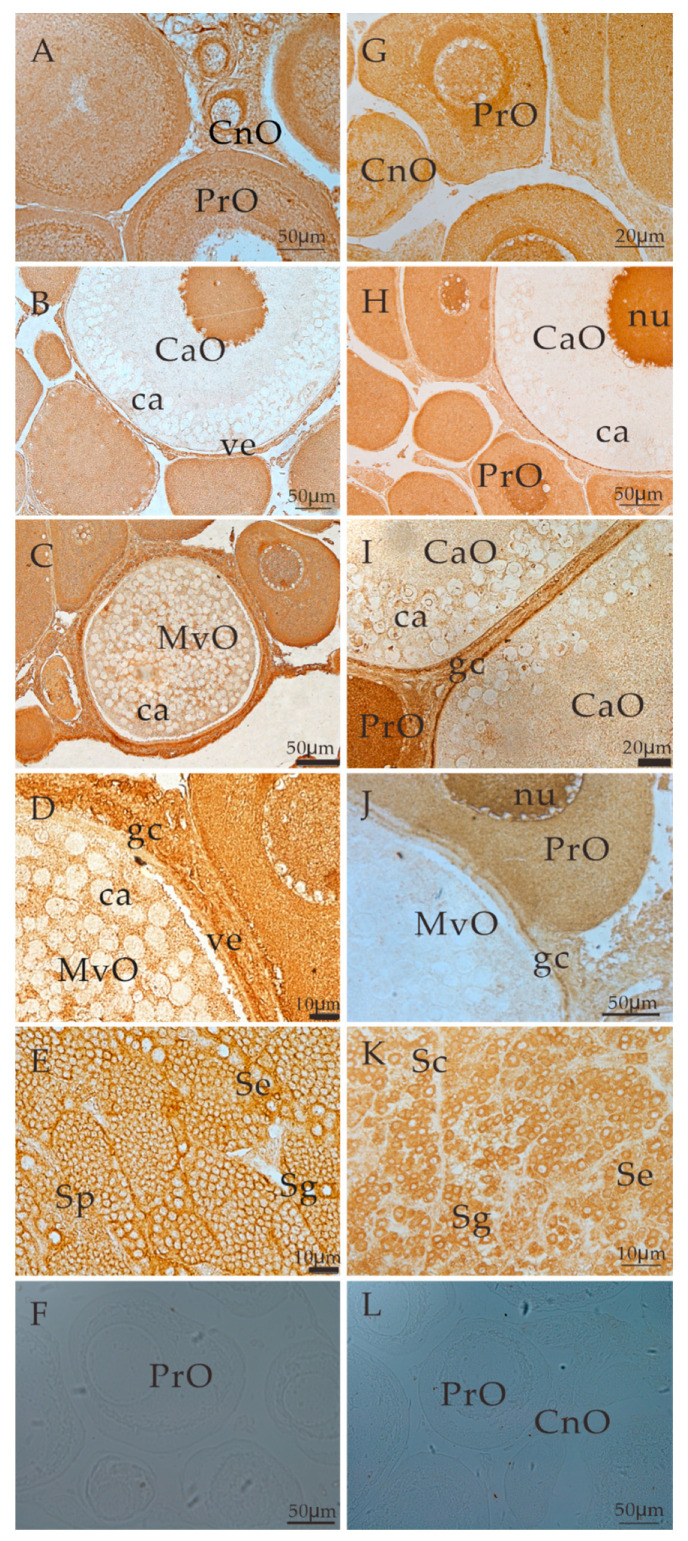
Localization of ALK5 (**A**–**F**) and BMPR2 (**G**–**L**) in the ovary and testis of *S. prenanti*. The sections of the ovary at the primary growth stage (**A**), the previtellogenic stage (**C**), and the mid- to the late-vitellogenic stage (**D**), and of the testis (**E**,**K**) were immunostained with anti-ALK5 (**A**–**E**) and BMPR2 (**G**–**K**) antiserum. Panel F and L: the ovary section immunostained with no antiserum. CnO, chromatin nucleolar oocytes; PrO, perinucleolar oocytes; CaO, cortical alveoli oocytes; MvO, mid-vitellogenic oocytes; nu, nucleolus; ve, vitelline envelope; ca, cortical alveoli; Sg, spermatogonia; Sp, spermatid; gc, granulosa cells; Se, Sertoli cells.

**Figure 5 animals-11-01365-f005:**
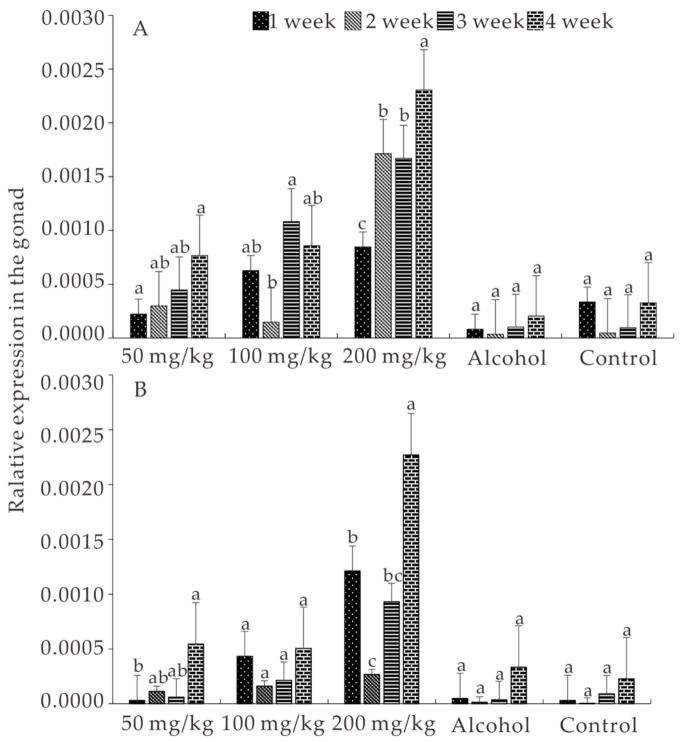
Concentration and time effects of *ALK5* (**A**) and *BMPR2* (**B**) expression in E_2_-fed *S. prenanti*. The data are represented as the means ± SEMs (n = 10). Means marked with different letters are significantly different (*p* < 0.05).

**Table 1 animals-11-01365-t001:** Sequences of oligonucleotide primers used in this study.

Primers	Sequence (5′-3′)	Use
*ALK5*-F1	GGACCATCGCCAGGACCATC	Sequence amplification
*ALK5*-F2	CTTCTACATCTGCCACAGC	
*ALK5*-R1	GCCTCRCAGCTCTGCC	
*ALK5*-R2	GTTRGCGTACCAGCACTC	
*BMPR2*-F1	AACGAACGCTCCATATAC	Sequence amplification
*BMPR2*-R1	TGGGAGTGCTCCATCAGG	
*BMPR2*-F2	CHSHRGGRGGYGGAGC	
*BMPR2*-F3	GTCCGCCACAACTACAACG	
RACE *ALK5*-F1	TCCGTGCCAACATCCCA	Sequence amplification
RACE *ALK5*-F2	GAGGGAGTGCTGGTACGCTA	
RACE *ALK5*-R1	CCTTCCCTATGCTCTCCTG	
RACE *ALK5*-R2	GATGATGGTCCTGGCGA	
*β-Actin* Q-F1	GATTCGCTGGAGATGATGCT	Real-time quantitative PCR analysis
*β-Actin* Q-R1	CGTTCTAGAAGGTGTGATGCC	
18S *r*RNA Q-F1	ACCACCCACAGAATCGAGAAA	Real-time quantitative PCR analysis
18S *r*RNA Q-R1	GCCTGCGGCTTAATTTGACT	
*ALK5* Q-F1	ACCCGAGGTGCTGGATGACT	Real-time quantitative PCR analysis
*ALK5* Q-R1	TTGGGATGTTGGCACGGAG	
*BMPR2* Q-F1	TGACGGGAAATCGCCTC	Real-time quantitative PCR analysis
*BMPR2* Q-R1	CTCAAAGGAGGGGTGGTTC	

Note: F, sense primer; R, antisense primer.

## Data Availability

Data will be available upon request.

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
