# Peer review of "Estradiol Upregulates the Expression of the TGF-β Receptors ALK5 and BMPR2 during the Gonadal Development of Schizothorax prenanti"

_animals, 2021, doi:10.3390/ani11051365_

Round 1
Reviewer 1 Report
Activin receptor-like kinase 5 (Alk5) and bone morphogenetic protein receptor 2 (Bmpr2) are essential factors in transmitting signals that regulate gonadal development in fish. After reviewing these factors, the authors leat the manuscript to an-depth exploration of the functions of alk5 and bmpr2 during the gonadal development of Schizothorax prenanti, an endemic fish species with great economic importance in aquaculture in Western China.
Results revealed that alk5 and bmpr2 might play a crucial role in both folliculogenesis and spermatogenesis in S. prenanti
As far as I can perceive, this is a well-designed experimental approach, and I only have minor observations
- Please provide more information about the zootechnical management of fish.
- Do you have any evidence that the amplification effectiveness of the designed primers is adequate?
- The quality of some figures in terms of font size (too little) and resolution is poor.
- If possible, I would recommend an illustration that emphasizes what was found in this experiment.
Author Response
Point-by-point responses to reviewer 2 comments/suggestions
1. The authors should unify the name of the genes. These must be written in lowercase and italic (lines 39, and others).
Response: We adjusted the abbreviation and caps in revised manuscript (Line 39-40; Line 327;Line 345, 346 and 348; Line 391 and 392).
2. Lines 60-66: in which species? Please, explain it.
Response: Lines 60-66 We had added the species in Lines 61-68.
3 . Line 95: I don’t understand this sentence.
Response: Line95 We revised “Tissues (n = 5), and tissues, including the hypothalamus, telencephalon, epithalamus...” as “Tissues (n = 5), including the hypothalamus, telencephalon, epithalamus…” (Line 98).
4. Line 189: Before to realize a parametric test, as ANOVA, the authors should confirm the normality and homoscedasticity of their data. In gene expression results, non-normal data are common, and often require a mathematical transformation, usually logarithmic, to normalize the data before statistical analysis.
Response: In the revised manuscript, all the data were re-analyzed by the One-Sample Kolmogorov-Smirnov Test of SPSS 21.0 software. These results showed that the data of gene expression were in accordance with the normality and homoscedasticity distribution, except the tissue distribution of alk5 and bmpr2 in S. prenanti determined by qRT-PCR. For re-analysis of those data, we have utilized the mathematical transformation (logarithmic 10+10) to normalize the data before statistical analysis. A detailed description of the method was added in the revised manuscript (Line 232-235).
5. Line 266: If the expression of these genes is higher in this group, p-value should be less than 0.05. Please, revise it.
Response: After reconfirmation the data, p-value has been modified in the revised manuscript. (Line 310).

Reviewer 2 Report
Dear authors,
This study analyses the expression and localization of Alk5 and Bmpr2 and its relation with oestradiol. The manuscript is well written and structured, the introduction provides sufficient background, the research design is appropriate, the results are clearly presented and the conclusions are supported by these results. However, minor changes are necessary:
- The authors should unify the name of the genes. These must be written in lowercase and italic (lines 39, and others).
- Lines 60-66: in which species? Please, explain it.
- Line 95: I don’t understand this sentence.
- Line 189: Before to realize a parametric test, as ANOVA, the authors should confirm the normality and homoscedasticity of their data. In gene expression results, non-normal data are common, and often require a mathematical transformation, usually logarithmic, to normalize the data before statistical analysis.
- Line 266: If the expression of these genes is higher in this group, p-value should be less than 0.05. Please, revise it.
I hope these recommendation improve the quality of manuscript.
Author Response
Point-by-point responses to reviewer 2 comments/suggestions
1.The authors should unify the name of the genes. These must be written in lowercase and italic (lines 39, and others).
Response: We adjusted the abbreviation and caps in revised manuscript (Line 39-40; Line 327;Line 345, 346 and 348; Line 391 and 392).
2. Lines 60-66: in which species? Please, explain it.
Response: Lines 60-66 We had added the species in Lines 61-68.
3 . Line 95: I don’t understand this sentence.
Response: Line95 We revised “Tissues (n = 5), and tissues, including the hypothalamus, telencephalon, epithalamus...” as “Tissues (n = 5), including the hypothalamus, telencephalon, epithalamus…” (Line 98).
4. Line 189: Before to realize a parametric test, as ANOVA, the authors should confirm the normality and homoscedasticity of their data. In gene expression results, non-normal data are common, and often require a mathematical transformation, usually logarithmic, to normalize the data before statistical analysis.
Response: In the revised manuscript, all the data were re-analyzed by the One-Sample Kolmogorov-Smirnov Test of SPSS 21.0 software. These results showed that the data of gene expression were in accordance with the normality and homoscedasticity distribution, except the tissue distribution of alk5 and bmpr2 in S. prenanti determined by qRT-PCR. For re-analysis of those data, we have utilized the mathematical transformation (logarithmic 10+10) to normalize the data before statistical analysis. A detailed description of the method was added in the revised manuscript (Line 232-235).
5. Line 266: If the expression of these genes is higher in this group, p-value should be less than 0.05. Please, revise it.
Response: After reconfirmation the data, p-value has been modified in the revised manuscript. (Line 310).
